# Neural Fixed-Point Acceleration for Convex Optimization

**Shobha Venkataraman**[*]
**Brandon Amos**[*]
*Facebook AI*

## Abstract

Fixed-point iterations are at the heart of numerical computing and are often a computational bottleneck in real-time applications, which typically instead need a fast solution of moderate accuracy. Classical acceleration methods for fixed-point problems focus on designing algorithms with theoretical guarantees that apply to *any* fixed-point problem. We present *neural fixed-point acceleration*, a framework to automatically learn to accelerate convex fixed-point problems that are drawn from a distribution, using ideas from meta-learning and classical acceleration algorithms. We apply our framework to SCS, the state-of-the-art solver for convex cone programming, and design models and loss functions to overcome the challenges of learning over unrolled optimization and acceleration instabilities. Our work brings neural acceleration into any optimization problem expressible with CVXPY. The source code behind this paper is available at github.com/facebookresearch/neural-scs.

## 1. Introduction

Continuous fixed-point problems are a computational primitive in numerical computing, optimization, machine learning, and the natural and social sciences. Given a map $f : \mathbb{R}^n \to \mathbb{R}^n$, a *fixed point* $x \in \mathbb{R}^n$ is where $f(x) = x$. *Fixed-point iterations* repeatedly apply $f$ until the solution is reached and provably converge under assumptions of $f$. Most solutions to optimization problems can be seen as finding a fixed point mapping of the iterates, *e.g.* in the convex setting, $f$ could step a primal-dual iterate closer to the KKT optimality conditions of the problem, which remains fixed once it is reached. Recently in the machine learning community, fixed point computations have been brought into the modeling pipeline through the use of differentiable convex optimization (Domke, 2012; Gould et al., 2016; Amos and Kolter, 2017; Agrawal et al., 2019; Lee et al., 2019), differentiable control (Amos et al., 2018), deep equilibrium models (Bai et al., 2019, 2020), and sinkhorn iterations (Mena et al., 2018).

Fixed-point computations are often a computational bottleneck in the larger systems they are a part of. *Accelerating* (*i.e.* speeding up) fixed point computations is an active area of optimization research that involves using the knowledge of prior iterates to improve the future ones. These improve over standard fixed-point iterations but are classically done without learning. The optimization community has traditionally not explored learned solvers because of the lack of theoretical guarantees on learned solvers. For many real-time applications, though, traditional fixed-point solvers can be too slow; instead we need a fast low-accuracy solution. Further, fixed-point problems repeatedly solved in an application typically share a lot of structure and so an application naturally induces a distribution of fixed-point *problem instances*. This raises the question: can we learn a fast and sufficiently-accurate fixed-point solver, when the problem instances are drawn from a fixed distribution?

---

[*]. Equal contribution.

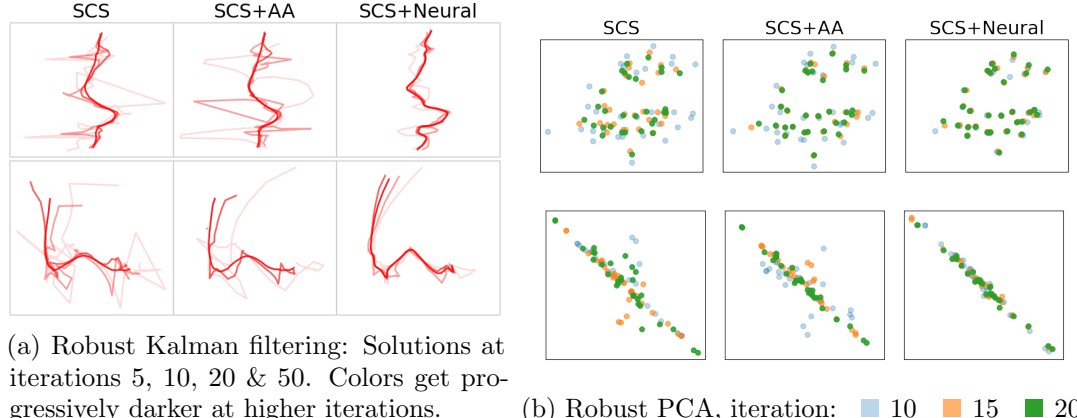

(a) Robust Kalman filtering: Solutions at iterations 5, 10, 20 & 50. Colors get progressively darker at higher iterations.

(b) Robust PCA, iteration: ▪ 10 ▪ 15 ▪ 20

Figure 1: Visualizing neural-accelerated test instances for robust Kalman filtering and robust PCA. Each line is a single instance. Neural-accelerated SCS quickly stabilizes to a solution, while the SCS and SCS+AA iterations exhibit higher variance.

In this paper, we explore the problem of learning to accelerate fixed-point problem instances drawn from a distribution, which we term *neural fixed-point acceleration*. We focus on convex optimization to ground our work in real applications, including real-time ones such as Tedrake et al. (2010); Mattingley and Boyd (2010). We design a framework for our problem based on *learning to optimize*, *i.e.*, meta-learning (Sect. 2): we learn a model that accelerates the fixed-point computations on a fixed distribution of problems, by repeatedly backpropagating through their unrolled computations. We build on ideas from classical acceleration: we learn a model that uses the prior iterates to improve them, for problems in this distribution. Our framework also captures classical acceleration methods as an instance.

We show how we can learn an acceleration model for convex cone programming with this framework. We focus on SCS (O'Donoghue et al., 2016), which is the state-of-the-art default cone solver in CVXPY (Diamond and Boyd, 2016). However, learning to optimize and acceleration are notoriously hard problems with instabilities and poor solutions, so there are challenges in applying our framework to SCS, which has complex fixed-point computations and interdependencies. Through careful design of models and loss functions, we address the challenges of differentiating through unrolled SCS computations and the subtleties of interweaving model updates with iterate history. Our experiments show that we consistently accelerate SCS in three applications – lasso, robust PCA and robust Kalman filtering.

## 2. Related work

**Learned optimizers and meta-learning.** The machine learning community has recently explored many approaches to learning to improve the solutions to optimization problems. These applications have wide-ranging applications, *e.g.* in optimal power flow (Baker, 2020; Donti et al., 2021), combinatorial optimization Khalil et al. (2016); Dai et al. (2017); Nair et al. (2020); Bengio et al. (2020), and differential equations (Li et al., 2020; Poli et al., 2020; Kochkov et al., 2021). The meta-learning and learning to optimize literature, *e.g.* (Li and Malik, 2016; Finn et al., 2017; Wichrowska et al., 2017; Andrychowicz et al., 2016; Metz et al., 2019, 2021; Gregor and LeCun, 2010), focuses on learning better solutions to

---

**Algorithm 1** Neural fixed-point acceleration augments standard fixed-point computations with a learned initialization and updates to the iterates.

---

**Inputs:** Context $\phi$, parameters $\theta$, and fixed-point map $f$.

$[x_1, h_1] = g_\theta^{\text{init}}(\phi)$          $\triangleright$ Initial hidden state and iterate

**for** fixed-point iteration $t = 1..T$ **do**

 $\tilde{x}_{t+1} = f(x_t; \phi)$         $\triangleright$ Original fixed-point iteration

 $x_{t+1}, h_{t+1} = g_\theta^{\text{acc}}(x_t, \tilde{x}_{t+1}, h_t)$      $\triangleright$ Acceleration

**end for**

---

parameter learning problems that arise for machine learning tasks. Our work is the most strongly connected to the learning to optimize work here and can be seen as an application of these methods to fixed-point computations and convex cone programming.

**Fixed-point problems and acceleration.** Accelerating fixed-point computations date back decades and include *Anderson Acceleration* (AA) (Anderson, 1965) and *Broyden's method* (Broyden, 1965), or variations such as Walker and Ni (2011); Zhang et al. (2020).

## 3. Neural fixed-point acceleration

### 3.1 Problem formulation

We are interested in settings and systems that involve solving a known distribution over fixed-point problems. Each fixed-point problem depends on a *context* $\phi \in \mathbb{R}^m$ that we have a distribution over $\mathcal{P}(\phi)$. The distribution $\mathcal{P}(\phi)$ induces a distribution over fixed-point problems $f(x; \phi) = x$ with a fixed-point map $f$ that depends on the context. Informally, our objective will be to solve this class of fixed-point problems as fast as possible. Notationally, other settings refer to $\phi$ as a "parameter" or "conditioning variable," but here we will consistently use "context." We next consider a general solver for fixed-point problems that captures classical acceleration methods as an instance, and can also be parameterized with some $\theta$ and learned to go beyond classical solvers. Given a fixed context $\phi$, we solve the fixed-point problem with alg. 1. At each time step $t$ we maintain the *fixed-point iterations* $x_t$ and a *hidden state* $h_t$. The *initializer* $g_\theta^{\text{init}}$ depends on the context $\phi$ provides the starting iterate and hidden state and the *acceleration* $g_\theta^{\text{acc}}$ updates the iterate after observing the application of the fixed-point map $f$.

**Proposition 1** *Alg. 1 captures Anderson Acceleration as stated* e.g., *in Zhang et al. (2020).*

This can be seen by making the hidden state a list of the previous $k$ fixed-point iterations, and there would be no parameters $\theta$. The initializer $g_\theta^{\text{init}}$ would return a deterministic, problem-specific initial iterate, and the acceleration $g_\theta^{\text{acc}}$ would apply the standard update and append the fixed-point iteration to the hidden state.

### 3.2 Modeling and optimization

We first parameterize the models behind the fixed-point updates in Alg. 1. In neural acceleration, we will use learned models for $g_\theta^{\text{init}}$ and $g_\theta^{\text{acc}}$. We experimentally found that

we achieve good results a standard MLP for $g_\theta^{\text{init}}$ and a recurrent model such as an LSTM (Hochreiter and Schmidhuber, 1997) or GRU (Cho et al., 2014) for $g_\theta^{\text{acc}}$. While the appropriate models vary by application, a recurrent structure is a particularly good fit as it encapsulates the history of iterates in the hidden state, and uses that to predict a future iterate.

Next, we define and optimize an objective for learning that characterizes how well the fixed-point iterations are solved. Here, we use the *fixed-point residual norms* defined by $\mathcal{R}(x; \phi) := ||x - f(x; \phi)||_2$. This is a natural choice for the objective as the convergence analysis of classical acceleration methods are built around the fixed-point residual. Our learning objective is thus to find the parameters to minimize the fixed-point residual norms in every iteration across the distribution of fixed-point problem instances, *i.e.*

$$\underset{\theta}{\text{minimize}} \ \mathbb{E}_{\phi \sim \mathcal{P}(\phi)} \sum_{t<T} \mathcal{R}(x_t; \phi) / \mathcal{R}_0(\phi), \tag{1}$$

where $T$ is the maximum number of iterations to apply and $\mathcal{R}_0$ is a normalization factor that is useful when the fixed-point residuals have different magnitudes. We optimize eq. (1) with gradient descent, which requires the derivatives of the fixed-point map $\nabla_x f(x)$.

## 4. Accelerating Convex Cone Programming

We have added neural acceleration to SCS (*Neural SCS*) and integrated it with CVXPY. SCS uses fixed-point iterations to solve cone programs in standard form:

$$\text{minimize} \ c^T x \ \text{ subject to } \ Ax + s = b, \quad (x, s) \in \mathbb{R}^n \times \mathcal{K}, \tag{2}$$

where $x \in \mathbb{R}^n$ is the primal variable, $s \in \mathbb{R}^m$ is the primal slack variable, $y \in \mathbb{R}^m$ is the dual variable, and $r \in \mathbb{R}^n$ is the dual residual. The set $\mathcal{K} \in \mathbb{R}^m$ is a non-empty convex cone. The fixed-point computations in SCS consists of a projection onto an affine subspace by solving a linear system followed by a projection onto the convex cone constraints.

### 4.1 Designing Neural SCS

We now describe how we design Neural SCS as a realization of Alg. 1 in three key steps: modeling, differentiating through SCS, and designing the objective.

**Modeling.** The input parameters $\theta$ come from the initializations of the neural networks that we train, $g_\theta^{\text{init}}$ and $g_\theta^{\text{acc}}$. To construct the input context $\phi$ for a problem instance, we convert the problem instance into its standard form (eq. (2)), and use the quantities $A, b$ and $c$, *i.e.* $\phi = [v(A); b; c]$ where $v : \mathbb{R}^{m \times n} \to \mathbb{R}^{mn}$ vectorizes the matrix $A$. We use an MLP for $g_\theta^{\text{init}}$, and a multi-layer LSTM or GRU for $g_\theta^{\text{acc}}$.

**Differentiating through SCS.** Optimizing the loss in eq. (1) requires that we differentiate through the fixed-point iterations of SCS: 1) For the *linear system solve.* We use implicit differentiation, *e.g.* as described in Barron and Poole (2016). Further, for differentiating through SCS, for a linear system $Qu = v$, we only need to obtain the derivative $\frac{\partial u}{\partial v}$, since the fixed-point computation repeatedly solves linear systems with the same $Q$, but different $v$. This also lets us use an LU decomposition of $Q$ to speed up the computation of the original linear system solve and its derivative. 2) for the *cone projections,* we use the derivatives from Ali et al. (2017); Busseti et al. (2019).

Table 1: Sizes of convex cone problems in standard form

|  | Lasso | PCA | Kalman Filter |
|---|---|---|---|
| Variables $n$ | 102 | 741 | 655 |
| Constraints $m$ | 204 | 832 | 852 |
| nonzeros in $A$ | 5204 | 1191 | 1652 |

| | | Lasso | PCA | Kalman Filter |
|---|---|---|---|---|
| Cone dims | Zero | 0 | 90 | 350 |
| | Non-negative | 100 | 181 | 100 |
| | PSD | none | [33] | none |
| | Second-order | [101, 3] | none | [102] + [3]×100 |

**Designing the Loss.** The natural choice for the learning objective is the fixed-point residual norm of SCS. With this objective, the interacting algorithmic components of SCS cause $g_\theta^{\mathrm{acc}}$ and $g_\theta^{\mathrm{init}}$ to learn poor models for the cone problem. In particular, SCS scales the iterates of feasible problems by $\tau$ for better conditioning. However, this causes a serious issue when optimizing the fixed-point residuals: shrinking the iterate-scaling $\tau$ artificially decreases the fixed-point residuals, allowing $g_\theta^{\mathrm{acc}}$ to have a good loss even with poor solutions.

We eliminate this issue by normalizing each $x_t$ by its corresponding $\tau$, similar to Busseti et al. (2019). Thus, the fixed-point residual norm becomes the $||x_t/\tau_t - f(x_t, \phi)/\tau_{f(x_t,\phi)}||$. We are then always measuring the residual norm with $\tau = 1$ for the learning objective, which does not modify the cone program that we are optimizing In addition, with this objective, we no longer need to learn or predict from $\tau$ in the models $g_\theta^{\mathrm{init}}$ and $g_\theta^{\mathrm{acc}}$.

## 4.2 Experiments

We demonstrate the experimental performance of SCS+Neural on 3 cone problems: Lasso (Tibshirani (1996)), Robust PCA (Candès et al. (2011)) and Robust Kalman Filtering, chosen similarly to O'Donoghue et al. (2016). Table 1 summarizes problem sizes, types of cones, and cone sizes used in our experiments. We use Adam (Kingma and Ba, 2014) to train for 100,000 model updates. We perform a hyperparameter sweep, and select models with the best validation loss in each problem class. For SCS+AA, we use the default history of 10 iterations. App. C.1.2 describes additional training details and the source code for our experiments is available online at github.com/facebookresearch/neural-scs.

**Results.** As an initial proof-of-concept, our experimental results focus on the number of iterations required to achieve required accuracy with SCS+Neural. Figure 2 shows the fixed-point, primal and dual residuals for SCS, SCS+AA, and SCS+Neural. It shows the mean and standard deviation of each residual per iteration, aggregated over all test instances for each solver. SCS+Neural consistently reaches a lower residual much faster than SCS or SCS+AA. e.g., in Lasso (fig. 2a) SCS+Neural reaches a fixed-point residual of 0.001 in 25 iterations, while SCS+AA and SCS take 35 and 50 iterations and SCS respectively. Our improvement for Kalman filtering (fig. 2c) is even higher: we reach a fixed-point residual of 0.01 in 5 iterations, compared to the 30 iterations taken by SCS and SCS+AA. In addition, SCS+AA consistently has high standard deviation, due to its well-known stability issues.

Improving the fixed-point residuals earlier also results in improving the primal/dual residuals earlier. For Robust PCA (fig. 2b), this improvement lasts throughout the 50 iterations. However, SCS+AA has a slight edge in the later iterations for Lasso and Kalman filtering, especially in the primal/dual residuals. These can be improved by adding a regularizer with the final primal-dual residuals to the loss (discussed in App. C.2.2).

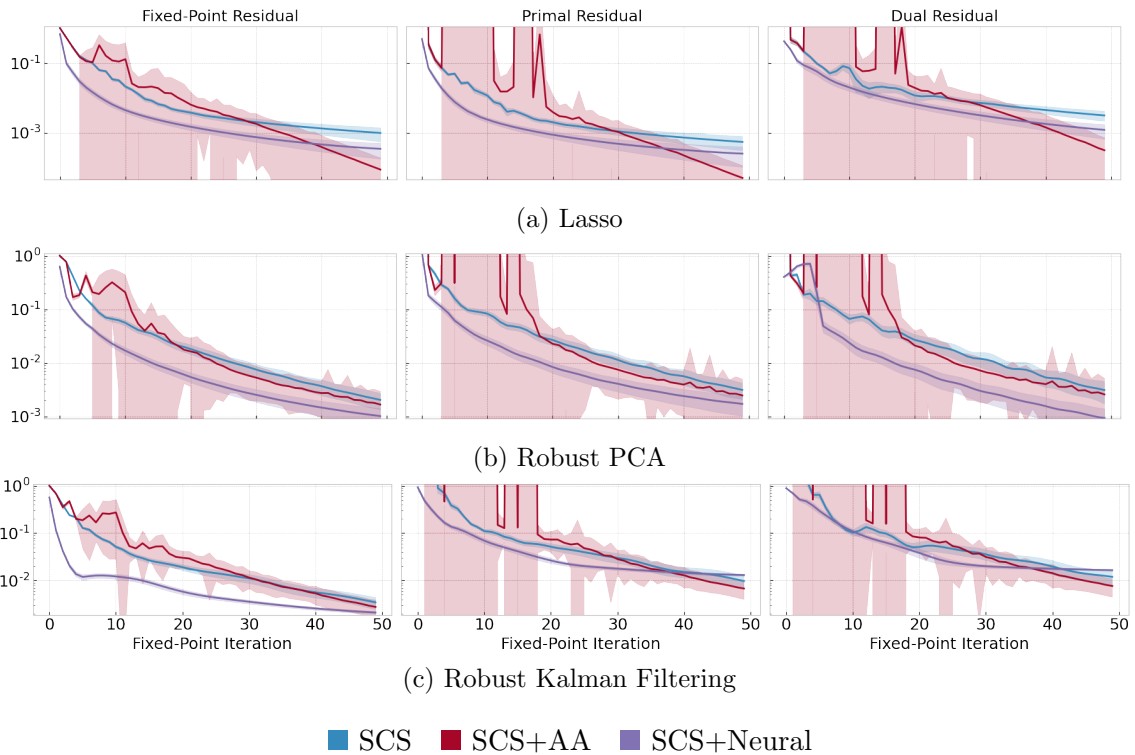

(a) Lasso

(b) Robust PCA

(c) Robust Kalman Filtering

■ SCS ■ SCS+AA ■ SCS+Neural

Figure 2: Neural accelerated SCS: Lasso, Robust PCA and Robust Kalman filtering

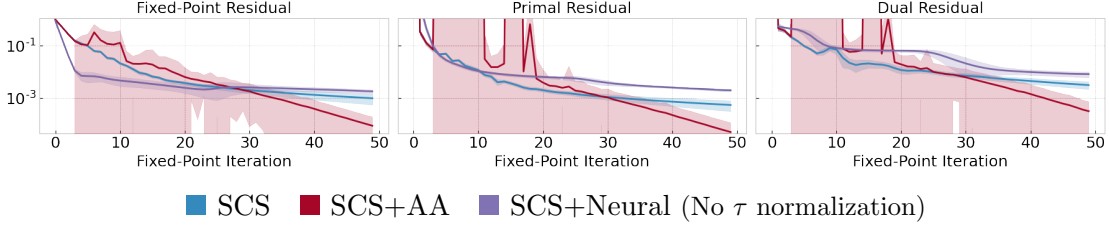

■ SCS ■ SCS+AA ■ SCS+Neural (No $\tau$ normalization)

Figure 3: Lasso without $\tau$ normalization: a failure mode of neural acceleration (that SCS+Neural overcomes with design).

*Importance of $\tau$ Normalization in Objective.* Figure 3 shows the residuals obtained for Lasso when SCS+Neural does not use $\tau$ normalization in the objective. The primal/dual residuals are worse than SCS and SCS+AA. The fixed-point residual shows an initial improvement, but finishes worse. As discussed in Sect. 4, this happens when SCS+Neural achieves a low loss by simply learning a low $\tau$, which we show in app. C.2.3.

## 5. Conclusion and future directions

We have demonstrated learned fixed-point acceleration for convex optimization. Future directions include scaling to larger convex optimization problems and accelerating fixed-point iterations in other domains, such as in motion planning (Mukadam et al., 2016), optimal transport (Mena et al., 2018), and deep equilibrium models (Bai et al., 2019, 2020).

## Acknowledgments

We thank Akshay Agrawal, Shane Barratt, Christian Kroer, and Alex Peysakhovich for insightful discussions and acknowledge the Python community (Van Rossum and Drake Jr, 1995; Oliphant, 2007) for developing the core set of tools that enabled this work, including PyTorch (Paszke et al., 2019), Hydra (Yadan, 2019), Jupyter (Kluyver et al., 2016), Matplotlib (Hunter, 2007), seaborn (Waskom et al., 2018), numpy (Oliphant, 2006; Van Der Walt et al., 2011), pandas (McKinney, 2012), and SciPy (Jones et al., 2014).

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

## Appendix A. Batched and Differentiable SCS

In this section, describe our batched and differentiable PyTorch implementation of SCS that enables the neural fixed-point acceleration as a software contribution.

We have implemented SCS in PyTorch, with support for the zero, non-negative, second-order, and positive semi-definite cones. Because our goal is to learn on multiple problem instances, we support batched version of SCS, so that we can solve a number of problem instances simultaneously. For this, we developed custom cone projection operators in PyTorch that allow us to perform batched differentiation.

SCS includes a number of enhancements in order to improve its speed and stability over a wide range of applications. Our implementation in PyTorch supports all enhancements that improve convergence, including scaling the problem data so that it is equilibrated, over-relaxation, and scaling the iterates between each fixed point iteration. Our implementation is thus fully-featured in its ability to achieve convergence using only as many fixed point iterations as SCS.

We are also able to achieve significant improvements in speed through the use of PyTorch JIT and a GPU. However, the focus of this work is on proof-of-concept of neural fixed-point acceleration, and so we have not yet optimized PyTorch-SCS for speed and scale. Our key limitation comes from the necessity of using dense operations in PyTorch, because PyTorch's functionality is primarily centered on dense tensors. While the cone programs are extremely sparse, we are unable to take advantage of its sparsity; this limits the scale of the problems that can be solved. We plan to address these limitations in a future implementation of a differentiable cone solver.

## Appendix B. Application: ISTA for elastic net regression

As a first simple application for demonstrating and grounding our fixed-point acceleration, we consider the elastic net regression setting that Zhang et al. (2020) uses to demonstrate the improved convergence of their Anderson Acceleration variant. This setting involves solving elastic net regression (Zou and Hastie, 2005) problems of the form

$$\text{minimize } \frac{1}{2}||Ax - b||_2^2 + \mu \left( \frac{1-\beta}{2}||x||_2^2 + \beta||x||_1 \right), \tag{3}$$

where $A \in \mathbb{R}^{m \times n}$, $b \in \mathbb{R}^m$. We refer to the objective here as $g(x)$. We solve this with the fixed point computations from the iterative shrinkage-thresholding algorithm

$$f(x) = S_{\alpha\mu/2} \left( x - \alpha \left( A^\top (Ax - b) + \frac{\mu}{2}x \right) \right), \tag{4}$$

with the shrinkage operator $S_\kappa(x)_i = \text{sign}(x_i)(|x_i| - \kappa)_+$. We follow the hyper-parameters and sampling procedures described in Zhang et al. (2020) and use their Anderson Acceleration with a lookback history of 5 iterations. We set $\mu = 0.001\mu_{\max}$, $\mu_{\max} = ||A^\top b||_\infty$, $\alpha = 1.8/L$, $L = (A^\top A) + \mu/2$, and $\beta = 1/2$. We take $m = n = 25$ and sample $A$ from a Gaussian, $\hat{x}$ from a sparse Gaussian with sparsity 0.1, and generate $b = A\hat{x} + 0.1w$, where $w$ is also sampled from a Gaussian.

We demonstrate in fig. 4 that we competitively accelerate these fixed-point computations. We do this using an MLP for the initialization, GRU for the recurrent unit. In addition

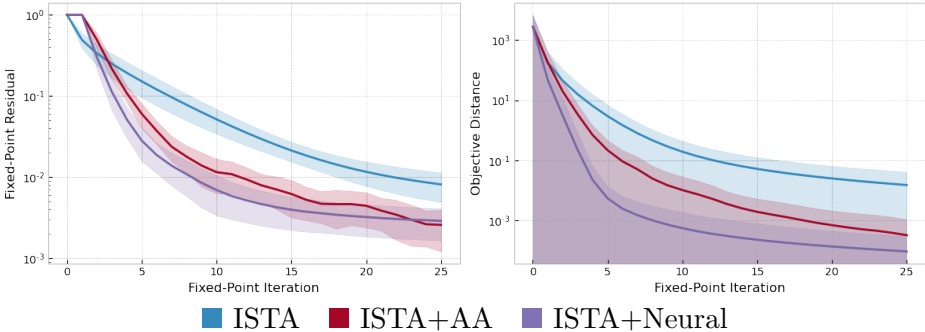

Figure 4: Learning to accelerate ISTA for solving elastic net regression problems. This shows the average fixed-point residual and distance to the optimal objective of a single training run averaged over 50 test samples.

to showing the training objective of the normalized fixed-point residuals $\mathcal{R}(x)/\mathcal{R}_0$, we also report the distance from the optimal objective $||g(x) - g(x^\star)||_2^2$, where we obtain $x^\star$ by using SCS to obtain a high-accuracy solution to eq. (4).

## Appendix C. Additional Experiments on Neural SCS

### C.1 Background and setup

In this section, we provide experimental setup details for results with SCS+Neural. We describe first the different cone problems we use, and then describe additional experimental setup details.

#### C.1.1 CONE PROGRAMS

**Lasso.** Lasso Tibshirani (1996) is a well-known machine learning problem formulated as follows:

$$\underset{z}{\text{minimize}} \ (1/2)||Fz - g||_2^2 + \mu||z||_1$$

where $z \in \mathbb{R}^p$, and where $F \in \mathbb{R}^{q \times p}$, $g \in \mathbb{R}^p$ and $\mu \in \mathbb{R}_+$ are data. In our experiments, we draw problem instances from the same distributions as O'Donoghue et al. (2016): we generate $F$ as $q \times p$ matrix with entries from $\mathcal{N}(0, 1)$; we then generate a sparse vector $z^*$ with entries from $\mathcal{N}(0, 1)$, and set a random 90% of its entries to 0; we compute $g = Fz^* + w$, where $w \sim \mathcal{N}(0, 0.1)$; we set $\mu = 0.1||F^T g||_\infty$. We use $p = 100$ and $q = 50$.

**Robust PCA.** Robust Principal Components Analysis Candès et al. (2011) recovers a low rank matrix of measurements that have been corrupted by sparse noise by solving

$$\begin{aligned}
\text{minimize} \ & ||L||_* \\
\text{subject to} \ & ||S||_1 \leq \mu \\
& L + S = M
\end{aligned}$$

where variable $L \in \mathbb{R}^{p \times q}$ is the original low-rank matrix, variable $S \in \mathbb{R}^{p \times q}$ is the noise matrix, and the data is $M \in \mathbb{R}^{p \times q}$ the matrix of measurements, and $\mu \in \mathbb{R}_+$ that constrains the corrupting noise term.

Again, we draw problem instances from the same distributions as O'Donoghue et al. (2016): we generate a random rank-$r$ matrix $L^*$, and a random sparse matrix $S^*$ with no more than 10% non-zero entries. We set $\mu = ||S^*||_1$, and $M = L^* + S^*$. We use $p = 30$, $q = 3$ and $r = 2$.

**Robust Kalman Filtering.** Our third example applies robust Kalman filtering to the problem of tracking a moving vehicle from noisy location data. We follow the modeling of Diamond and Boyd as a linear dynamical system. To describe the problem, we introduce some notation: let $x_t \in \mathbb{R}^n$ denote the state at time $t \in \{0 \ldots T - 1\}$, and $y_t \in \mathbb{R}^r$ be the state measurement The dynamics of the system are denoted by matrices: $A$ as the drift matrix, $B$ as the input matrix and $C$ the observation matrix. We also allow for noise $v_t \in \mathbb{R}^r$, and input to the dynamical system $w_t \in \mathbb{R}^m$. With this, the problem model becomes:

$$
\begin{aligned}
\text{minimize} \quad & \Sigma_{t=0}^{N-1}(||w||_2^2 + \mu\psi_\rho(v_t)) \\
\text{s.t.} \quad & x_{t+1} = Ax_t + Bw_t, \quad t \in [0 \ldots T - 1] \\
& y_t = Cx_t + v_t, \quad t \in [0 \ldots T - 1]
\end{aligned}
$$

where our goal is to recover $x_t$ for all $t$, and where $\psi_\rho$ is the Huber function:

$$
\psi_\rho(a) = \begin{cases} ||a||_2 & ||a||_2 \leq \rho \\ 2\rho||a||_2 - \rho^2 & ||a||_2 \geq \rho \end{cases}
$$

We set up our dynamics matrices as in Diamond and Boyd, with $n = 50$ and $T = 12$. We generate $w_t^* \sim \mathcal{N}(0, 1)$, and initialize $x_0^*$ to be $\mathbf{0}$, and set $\mu$ and $\rho$ both to 2. We also generate noise $v_t^* \sim \mathcal{N}(0, 1)$, but increase $v_t^*$ by a factor of 20 for a randomly selected 20% time intervals. We simulate the system forward in time to obtain $x_t^*$ and $y_t$ for $T$ time steps. Table 1 summarizes the problem instances.

### C.1.2 Experimental Setup: Additional Details

For each problem, we create a training set of 100,000 problem instances (50,000 for Kalman filtering), and validation and test sets of 512 problem instances each (500 for Kalman filtering). We allow each problem instance to perform 50 fixed-point iterations for both training and evaluation. We perform a hyperparameter sweep across the parameters of the model, Adam, and training setup, which we detail in Table 2.

## C.2 Additional Results.

### C.2.1 Ablations

We present ablations that highlight the importance of the different pieces of SCS+Neural, using Lasso as a case study.

*Initializer.* Our first ablation examines the importance of the learned initializer $g_\theta^{\text{init}}(\phi)$ and the initial iterate and hidden state that it provides. We modify $g_\theta^{\text{init}}$ to output four possibilities: (1) neither initial iterate nor hidden state, (2) only the initial hidden state $h_1$, (3) only the initial iterate $x_1$, and (4) both the initial iterate and hidden state $[x_1, h_1]$. Note that in Case (1), the initial context $\phi$ is not used by the neural acceleration, while Case (4) matches alg. 1.

Table 2: Parameters used for hyperparameter sweep of SCS+Neural

| Adam | |
| --- | --- |
| learning rate | $[10^{-4}, 10^{-2}]$ |
| $\beta_1$ | 0.1, 0.5, 0.7, 0.9 |
| v $\beta_2$ | 0.1, 0.5, 0.7, 0.9, 0.99, 0.999 |
| cosine learning rate decay | True, False |
| **Neural Model** | |
| - use initial hidden state | True, False |
| - use initial iterate | True, False |
| Initializer: | |
| - hidden units | 128, 256, 512, 1024 |
| - activation function | relu, tanh, elu |
| - depth | $[0 \dots 4]$ |
| Encoder: | |
| - hidden units | 128, 256, 512, 1024 |
| - activation function | relu, tanh, elu |
| - depth | $[0 \dots 4]$ |
| Decoder: | |
| - hidden units | 128, 256, 512, 1024 |
| - activation function | relu, tanh, elu |
| - depth | $[0 \dots 4]$ |
| - weight scaling | $[2.0, 128.0]$ |
| Recurrent Cell: | |
| - model | LSTM, GRU |
| - hidden units | 128, 256, 512, 1024 |
| - depth | $[1 \dots 4]$ |
| **Misc** | |
| max gradient for clipping | $[10.0, 100.0]$ |
| batch size | 16, 32, 64, 128 [Lasso & PCA] |
| | 5, 10, 25, 50 [Kalman filter] |

Figure 5 shows the results for the four cases of $g_\theta^{\text{init}}$ in SCS+Neural, along with SCS and SCS+AA for comparison. They show the mean of all the test instances per iteration, averaged across three runs with different seeds. First, all four cases of SCS+Neural improve significantly over SCS and SCS+AA in the first 10 iterations, and are near-identical for the first 5-7 iterations. Further, two of the cases, i.e., Case (1) (where $g_\theta^{\text{init}}$ does not output anything), and Case (2) (where it only outputs $h_1$) show significantly less improvement than the other two cases; they are also near-identical. In addition, Case (3) (where $g_\theta^{\text{init}}$ outputs just the initial iterate $x_1$) is also near-identical to Case (4) (where it outputs both $[x_1, h_1]$). This suggests that $g_\theta^{\text{init}}$ able to start the fixed-point iteration with a good $x_1$, while the initial $h_1$ it has learned does not have much impact.

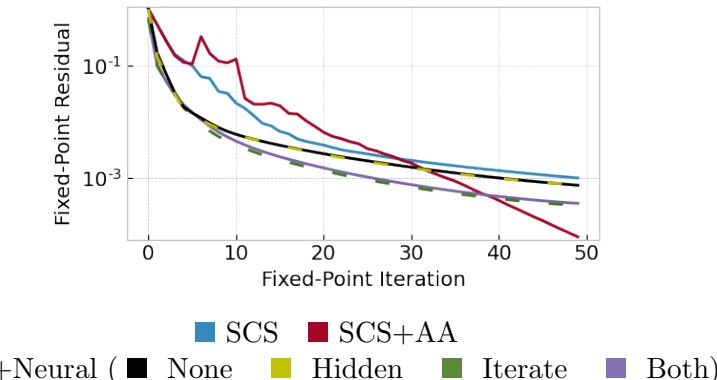

Figure 5: Initializer ablations: Lasso

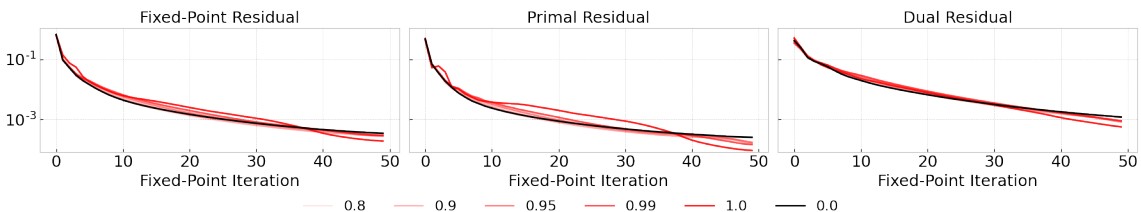

Figure 6: Ablations for regularization with primal & dual residuals: Lasso

### C.2.2 REGULARIZING WITH PRIMAL-DUAL RESIDUALS.

We can also optimize other losses beyond the fixed-point residuals to reflect more properties that we want our fixed-point solutions to have. Here we discuss how we can add the primal and dual residuals to the loss, which are different quantities that the fixed-point residuals. The loss is designed to minimize the fixed-point residual as early as possible, so sometimes, we see that the final primal-dual residuals of SCS+Neural are slightly worse than SCS and SCS+AA.

Because the primal/dual residuals also converge under the fixed-point map, we can adapt the loss to include them primal/dual residuals as well, *i.e.*, similar to eq. (1), we can define an updated learning objective:

$$\underset{\theta}{\text{minimize}} \ \mathbb{E}_{\phi \sim \mathcal{P}(\phi)}(1-\lambda) \sum_{t<T} \mathcal{R}(x_t; \phi)/\mathcal{R}_0(\phi) + \lambda ||[p(x_T, \phi); d(x_T, \phi)]||_2 \qquad (5)$$

where $\lambda \in [0,1]$ is the regularization parameter, $p$ and $d$ are the primal and dual residuals at $x_T$. At $\lambda = 0$, this is our original objective eq. (1); at $\lambda = 1$, this objective ignores the fixed-point residuals and only minimizes the final primal and dual residuals obtained after $T$ iterations. We ablate $\lambda$ in our experiments.

Our next ablation examines the impact of regularizing the loss with the final primal/dual residuals. Figure 6 shows all three residuals for SCS+Neural for $\lambda$ ranging from 0.8 to 1.0, in addition to the original SCS+Neural (with $\lambda = 0$). We only focus on high $\lambda$ because we see only marginal differences from the original SCS+Neural at lower $\lambda$. For clarity, we show only the means over all test instances for all seeds; the standard deviations are similar

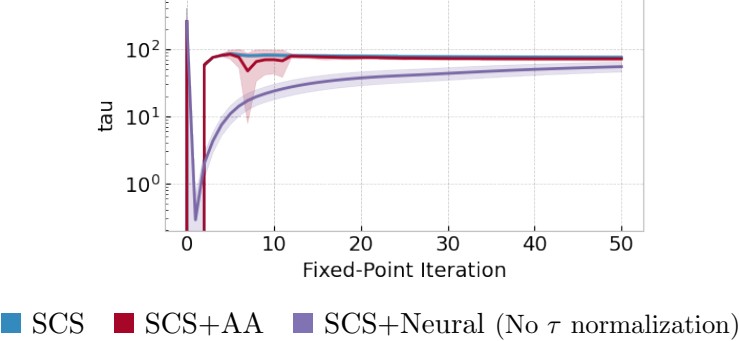

SCS ■  SCS+AA ■  SCS+Neural (No $\tau$ normalization)

Figure 7: We observe that without $\tau$ normalization, a failure mode of neural acceleration is that it learns to produce low $\tau$ values that artificially reduce the fixed-point residuals while not nicely solving the optimization problem.

to the earlier Lasso experiments. As $\lambda$ increases, all three residuals get a little worse than the original SCS+Neural in early iterations, while there is an improvement in all three residuals in the later iterations (past iteration 35). The maximum improvement in the final primal and dual residuals at $\lambda = 1$, when the learning objective is to minimize only the final primal/dual residuals. These results suggest that this regularization could be used to provide a flexible tradeoff of the residuals of the final solution for the speed of convergence of the fixed-point iteration.

### C.2.3 $\tau$ NORMALIZATION

We can understand the behavior of $g_\theta^{\mathrm{acc}}$ by examining how $\tau$ changes over the fixed-point iterations. Figure 7 shows the mean and standard deviation of the learned $\tau$ values, averaged across all test instances and across runs with all seeds. Note that SCS and SCS+AA quickly find their $\tau$ (by iteration 3-4), and deviate very little from it. SCS+Neural, however, starts at a very low $\tau$ that slowly increases; this results in very low initial fixed-point residuals (and thus a better loss for $g_\theta^{\mathrm{acc}}$), but poor quality solutions with high primal/dual residuals.

### C.2.4 VISUALIZING CONVERGENCE

Lastly, we discuss in more detail the visualizations of convergence that we illustrated in Sect. 1. Figure 1a shows the solutions of two different test instances for Robust Kalman filtering at iterations 5, 10, 20 and 50. Lighter paths show earlier iterations, and darker paths show later iterations. For both instances, e see that SCS+Neural has few visible light (intermediate) paths; most of them are covered by the final dark path, and those that are visible are of the lightest shade. This indicates that SCS+Neural has mostly converged by iteration 5, unlike SCS and SCS+AA, which have many more visible intermediate (light) paths around their final path. Figure 1b shows the solutions for two instances in Robust PCA at iterations 10, 15 and 20 for SCS, SCS+AA and SCS+Neural. It is clear that, for both instances, the SCS+Neural has almost converged by iteration 10. In contrast, SCS and SCS+AA show many more visible distinct points at iterations 10 and 15, indicating they have not yet converged.

