# OpenReview forum: "Neural Fixed-Point Acceleration for Convex Optimization"
_ICML.cc/2021/Workshop/AutoML — AutoML@ICML2021 Poster_

### Official Review · Reviewer_nJcA · 2021-06-11
**Interesting approach to improve fixed-point methods in a data driven way**

**Rating:** 8
**Confidence:** 4

**Review:**

The authors consider the problem of finding fixed points, which is essential for many optimization problems. The authors argue that typically methods for solving fixed-point problems are designed for having theoretical guarantees. However, in practice often methods are preferred that yield decent solutions as fast as possible.
Thus, the authors propose to adapt the learning to optimize framework to acceleration fixed-point methods by learning the initial point as well as an accelerator. The authors use two neural networks for this purpose, which parameters are learned by differentiating through the fixed-point iterations (as such, it is assumed that the fixed-point map is differentiable).

The proposed framework is evaluated on solving different classes convex cone programs (Lasso, Robust PCA, Robust Kalman Filtering) in CVXPY by improving the default optimizer (SCS) with the learned initializer and accelerator. The proposed method outperforms the baselines in most experiments.

While I wouldn't consider this work a typical AutoML paper, it goes in the direction of algorithm configuration and employs methods from the meta learning community and thus is a valuable contribution to this workshop. The paper is overall well written, presents an interesting idea and empirical evaluations.

---

### Official Review · Reviewer_g8c4 · 2021-06-13
**The proposed work is interesting, and of great value, yet, some claims need justification and more elaboration.**

**Rating:** 6
**Confidence:** 3

**Review:**

This work introduces an implementation of using meta-learning to accelerate the fixed-point iterations computation. Basically, it augments standard fixed-point computations with a learned initialization and updates to the iterates based on MLP and LSTM approaches. To show the effectiveness of the proposed work, it is used to solve 3 convex cone problems: Lasso, Robust PCA and Robust Kalman Filtering, where it did reduce the number of iterations used in the fixed-point calculations. The paper has the following pros and cons:

Pros:
1) In general, the proposed neural fixed-point acceleration seems simple, yet, interesting and of a great value for many numerical analysis applications that require fast performance.
2) This is the first learned work for accelerating fixed-point iterations computation. So, it is of high originality.
3) The writeup is good, easy-to-file, and contains a lot of useful appendices.
4) Experimental evaluation shows superiority of the proposed work over the traditional implementation. It can achieve 50% reduction in the number of iterations.

Cons:
1) The related work part was very shallow and needs to be more comprehensive.
2) The selection of using MLP and LSTM was not justified beyond experimentally. It could be the case that other methods and neural architecture work better with fixed-point problems of certain distributions. Is there any heuristics or analysis regarding that part?

---

### Official Review · Reviewer_NtbB · 2021-06-18
**Nice approach to accelerating fixed-point iterations. Practicality remains an issue, and robustness should be explored.**

**Rating:** 7
**Confidence:** 3

**Review:**

This paper proposes to use neural networks to accelerate the convergence of fixed-point iterations when the fixed-point mapping is drawn from a distribution over mappings (parameterized by a "context"). As a concrete example, the paper derives a neural-accelerated SCS algorithm and uses it on 3 convex cone problems. The approach generally favors well and demonstrates generally faster convergence and improved robustness over the baselines.

I like the idea of using neural networks to accelerate fixed point iterations. I've seen plenty of examples of using neural networks to imitate optimization algorithms, but I don't recall seeing one used in service of acceleration. As acknowledged in the paper though, while the number of iterations can be reduced, the wall-clock time is likely longer due to the cost of running the neural network. The time should also be reported for transparency.

I'm also concerned about robustness to perturbation. How far out-of-distribution can we go before the approach hurts more than it helps? Wall-clock time aside, this approach seems most useful when you solve similar cone problems again and again. Is there a motivating example where this is the case? This would at the very least be a requirement in order to collect enough training data. In the appendix, I see that the network was trained on tens of thousands of problem instances, so there should be a motivating example that would facilitate this kind of training data. How does the approach perform when only trained on a handful or problem instances?

Overall I think this direction is worth pursuing further, even if there remain practical issues with the current setup.

---

### Decision · Program_Chairs · 2021-06-21

Accept (Poster)